# Gut Microbiota Associated with Gestational Health Conditions in a Sample of Mexican Women

**DOI:** 10.3390/nu14224818

**Published:** 2022-11-14

**Authors:** Tizziani Benítez-Guerrero, Juan Manuel Vélez-Ixta, Carmen Josefina Juárez-Castelán, Karina Corona-Cervantes, Alberto Piña-Escobedo, Helga Martínez-Corona, Amapola De Sales-Millán, Yair Cruz-Narváez, Carlos Yamel Gómez-Cruz, Tito Ramírez-Lozada, Gustavo Acosta-Altamirano, Mónica Sierra-Martínez, Paola Berenice Zárate-Segura, Jaime García-Mena

**Affiliations:** 1Departamento de Genética y Biología Molecular, Cinvestav, Av. Instituto Politécnico Nacional 2508, Ciudad de México 07360, Mexico; 2Laboratorio de Posgrado de Operaciones Unitarias, Escuela Superior de Ingeniería Química e Industrias Extractivas, Instituto Politécnico Nacional, Ciudad de México 07738, Mexico; 3Unidad de Ginecología y Obstetricia, Hospital Regional de Alta Especialidad de Ixtapaluca, Carretera Federal México-Puebla Km. 34.5, Col. Zoquiapan, Ixtapaluca 56530, Mexico; 4Dirección de Planeación, Enseñanza e Investigación, Hospital Regional de Alta Especialidad de Ixtapaluca, Carretera Federal México-Puebla Km. 34.5, Col. Zoquiapan, Ixtapaluca 56530, Mexico; 5Unidad de Investigación en Salud, Hospital Regional de Alta Especialidad de Ixtapaluca, Carretera Federal México-Puebla Km. 34.5, Col. Zoquiapan, Ixtapaluca 56530, Mexico; 6Laboratorio de Medicina Traslacional, Escuela Superior de Medicina, Instituto Politécnico Nacional, Ciudad de México 11340, Mexico

**Keywords:** mother feces, high-throughput DNA sequencing, fecal microbiota, gut microbiota, gestational diabetes, pre-eclampsia, pre-gestational diabetes, Bruker Solarix XR, FT ICR MS

## Abstract

Gestational diabetes (GD), pre-gestational diabetes (PD), and pre-eclampsia (PE) are morbidities affecting gestational health which have been associated with dysbiosis of the mother’s gut microbiota. This study aimed to assess the extent of change in the gut microbiota diversity, short-chain fatty acids (SCFA) production, and fecal metabolites profile in a sample of Mexican women affected by these disorders. Fecal samples were collected from women with GD, PD, or PE in the third trimester of pregnancy, along with clinical and biochemical data. Gut microbiota was characterized by high-throughput DNA sequencing of V3-16S rRNA gene libraries; SCFA and metabolites were measured by High-Pressure Liquid Chromatography (HPLC) and (Fourier Transform Ion Cyclotron Mass Spectrometry (FT-ICR MS), respectively, in extracts prepared from feces. Although the results for fecal microbiota did not show statistically significant differences in alfa diversity for GD, PD, and PE concerning controls, there was a difference in beta diversity for GD versus CO, and a high abundance of Proteobacteria, followed by Firmicutes and Bacteroidota among gestational health conditions. DESeq2 analysis revealed bacterial genera associated with each health condition; the Spearman’s correlation analyses showed selected anthropometric, biochemical, dietary, and SCFA metadata associated with specific bacterial abundances, and although the HPLC did not show relevant differences in SCFA content among the studied groups, FT-ICR MS disclosed the presence of interesting metabolites of complex phenolic, valeric, arachidic, and caprylic acid nature. The major conclusion of our work is that GD, PD, and PE are associated with fecal bacterial microbiota profiles, with distinct predictive metagenomes.

## 1. Introduction

Under normal conditions, the gut microbiota establishes intricate and diverse symbiotic interactions with its host, being able to regulate immune responses, produce beneficial bioactive compounds for the host, i.e., vitamins, short-chain fatty acids (SCFA) [1], and contribute to other primary homeostasis processes [2]. The functional contribution of the microbiota is important as many chronic human diseases, including obesity [3], hypertension [4], cardiovascular disease [5], endothelial dysfunction [6], type 2 diabetes mellitus (T2DM) [7], gestational diabetes (GD) [8], and pre-eclampsia [9], have been associated with alterations in the diversity of gut microbial communities [10].

The American Diabetes Association defines GD as an abnormal blood glucose rise during the second or third trimester of pregnancy without any previous diabetes record [11]. The development of GD is associated with insulin secretion failure in a chronic insulin resistance context and with deficient glucose uptake in β-cells [12]. GD is diagnosed by the first recognition of hyperglycemia or impaired glucose tolerance during pregnancy through an oral glucose tolerance test (OGTT) [11].

The prevalence of GD has increased considerably, and it has become a global public health concern affecting 9.3–25.5% of pregnancies worldwide [13]. In Mexico, there is insufficient epidemiological information to assess GD prevalence, since there is no consensus to establish an accurate clinical diagnosis. It has been estimated that 8.7–17.7% of Mexican pregnant women develop GD [14], whereas the International Association of Diabetes and Pregnancy Study Groups estimated up to 30% [15]. Moreover, in clinical practice, only half of GD women are correctly diagnosed [16,17].

Risk factors associated with GD, such as being overweight, suffering from obesity, and hypertension, are highly prevalent in the Mexican population [18]. Other risk factors include maternal age greater than or equal to 35 years, multiparity, excessive weight gain or obesity during pregnancy, a GD history, first-degree relatives with diabetes, previous fetal macrosomia pregnancies [19], polycystic ovary syndrome, hypothyroidism [20], and diet [21].

Women with GD have an increased risk of comorbidities, thus leading to short and long-term poor life quality [22]. Associated complications comprise pre-eclampsia, postpartum infection, antenatal depression [23], metabolic syndrome [20], and cardiovascular diseases [24]. After delivery, women who suffer from GD are up to seven times at greater risk of T2DM development [25]. In addition to the maternal illness, neonates from mothers with GD have an increased risk of macrosomia, diabetic fetopathy, and neonatal hyperinsulinemia [26], and have a substantial risk for obesity and even T2DM later in life, contributing to the already growing diabetes epidemic [13,27].

It is important to mention, that gut microbiota change through pregnancy. This change mainly consists of an overall increase in Proteobacteria and Actinobacteria and reduced microbial richness in the third trimester [28]. In addition to these natural changes in the microbiota, an aberrant gut microbiota has been documented in GD individuals compared to healthy counterparts [29]. For instance, Danish women suffering from GD, in the third trimester of pregnancy, have reported persistent alterations in gut microbiota up to eight months after delivery, including high *Collinsella* abundance, reduction of SCFA-producing bacteria such as *Faecalibacterium* and *Bacteroides*, and reduction of *Isobaculum* [30]. Moreover, *Blautia* species, which are abundant in GD individuals, were correlated to an unhealthy metabolic profile. On the other hand, there was a *Ruminococcus* abundance reduction in postpartum GD women. A lower abundance of *Akkermansia* was also reported in women with gestational diabetes [30].

As a result of gut microbiota–host interactions studies in the third trimester of pregnancy, multiple mechanisms have been proposed for gut microbiota involvement in GD pathophysiology [29]. One of the most significant proposals is the role of SCFA. It has been reported that a reduction in the relative abundance of SCFA-producing bacteria is associated with an increase in blood glucose levels [26]. SCFA are involved in the activation of several important receptors, such as the peroxisome proliferator-activated receptor (PPAR), which helps to reduce the expression of inflammatory markers and oxidative stress [26]. In addition, SCFA interact with G protein-coupled receptors (GPR) to promote anorexic hormones such as glugacon-like peptide 1 (GLP-1) and peptide tyrosine tyrosine (PYY), thereby stimulating insulin secretion, promoting glucose metabolism, and inducing satiety [31].

Due to the high prevalence of GD and other gestational health conditions, such as pre-eclampsia and pre-gestational diabetes in Mexican women, it is of important interest to explore the gut microbiota and its produced metabolites to determine its relationship with the clinical, anthropometric, and dietary parameters in Mexican pregnant women diagnosed with GD and other morbidities during gestation.

## 2. Materials and Methods

### 2.1. Study Type and Selection of Subjects

An observational, retrospective, case-control study was conducted with the participation of attending Mexican pregnant women at the “Hospital Regional de Alta Especialidad de Ixtapaluca”, a governmental third-level hospital located in the State of Mexico (19°19′07″ N 98°52′56″ W). Fifty-four pregnant women in the third trimester of gestation were recruited and divided into four experimental groups: 30 healthy pregnant women (controls, CO), 11 pregnant women diagnosed with gestational diabetes (GD), 8 pregnant women diagnosed with preeclampsia (PE), and 5 women with a pre-pregnancy diagnosis of type 1 or 2 diabetes mellitus (PD). GD was diagnosed with the following criteria: fasting blood glucose ≥ 92 mg/dL, plasma glucose at 1 h post-stimulation with 75 g of anhydrous glucose ≥ 180 mg/dL, and plasma glucose at 2 h post-stimulation with 75 g of anhydrous glucose ≥ 153 mg/dL. For pre-eclampsia, the diagnosis was made after week 20, considering a systolic blood pressure ≥ 140 and a diastolic pressure ≥ 90, on more than 2 occasions with 4 h difference in a day, in addition to the presence of proteinuria. Pre-gestational diabetes patients were previously diagnosed before pregnancy with type 1 or 2 diabetes mellitus. Inclusion criteria were patients over 18 years, without associated pathologies in the case of the control group, no consumption of probiotics or antibiotics in the 3 months before the sample collection, and no gastrointestinal disease. The study was approved by the hospital’s Ethics Committee in Research (Comité de Ética en Investigación, CEI), register number NR-CEI-HRAEI-07-2021, and Research Committee (Comité de Investigación, CI), register number NR-07-2021. All participants consented to the collection of data and signed informed consent following the Declaration of Helsinki. It is important to mention that all sample collection occurred from July to October 2021, during the severe COVID-19 pandemic in Mexico, which restricted all the research operations in hospitals.

### 2.2. Data and Specimen Collection

Stool samples were provided by the participants who signed the informed consent and who met the inclusion criteria. Samples were stored at −70 °C until further use. Clinical parameters were obtained from each patient (age, parity, first-degree relatives diagnosed with diabetes mellitus, history of abortion, previous pregnancy with GD, fetal macrosomia in a previous pregnancy, etc.), anthropometric (weight, height, body mass index (BMI)), and metabolic (fasting glucose, glucose at 2 h after stimulation with 75 g of anhydrous glucose, glycosylated hemoglobin (HbA1c), cholesterol, and triglycerides). All data related to the diagnosis, gynecological-obstetric history, and risk factors were obtained from the clinical record. For each patient, a food frequency questionnaire, designed to obtain information about eating habits, was applied.

### 2.3. DNA Extraction

To perform the DNA extraction, 200 mg of fecal samples were processed using the FavorPrep ™ Stool DNA Isolation Mini Kit (Cat. FASTI 001-1, FAVORGEN© Biotech Corporation, Zhunan, Taiwan) following the manufacturer instructions. Subsequently, the integrity of the DNA fragments was confirmed by 0.5% agarose electrophoresis gel (90 V per 50 min) and the purity was assessed with the absorbance ratio 260/280 and 260/230 measured in the NanoDrop Lite Spectrophotometer (Thermo Scientific, Waltham, MA, USA) equipment.

### 2.4. Amplification of the V3 Region of the Bacterial 16S rRNA Gene

The fecal microbiota composition of experimental groups was established by sequencing the polymorphic region V3 of the bacterial 16S rRNA gene in each sample. Forward (V3-341F) and reverse (V3-518R) primers complementary to the upstream and downstream regions of the locus of interest were used [6]. Forward primer contains a known sequence barcode allowing individual sequences identification of samples in the pool. This procedure was performed by endpoint PCR. An amplicon of 281 bp was obtained, under the following amplification cycle: 3 min at 98 °C; 25 cycles (12 s to 98 °C, 15 s to 62 °C, and 10 s to 72 °C); and 5 min at 72 °C. The PCR product was visualized in 2% agarose gels. The amount of each amplicon was estimated by densitometry, using the Image Lab v.4.1 program, and a final library was made by mixing equal amounts of amplicons.

### 2.5. High-Throughput DNA Sequencing

The final library was purified using a highly sensitive 2% agarose gel stained with SYBR GOLD DNA (E-Gel™ EX, 2%, Invitrogen™, Cat. G401002, Waltham, MA, USA). The DNA library concentration and final size fragment were measured with 2100 Bioanalyzer Instrument (Agilent Technologies, Santa Clara, CA, USA) fragment analyzer, the resulting average size of the library was 263 bp. PCR emulsion was carried out using Ion OneTouchTM 200 Template Kit v2 DL (Life Technologies, Carlsbad, CA, USA), according to the manufacturer’s instructions. Enrichment of the amplicon with ionic spheres was carried out using Ion OneTouch ES (Life Technologies, Carlsbad, CA, USA). Sequencing was performed using the Ion 318 Kit V2 Chip (Cat. 4488146, Life Technologies, Carlsbad, CA, USA) and the Ion Torrent PGM system v4.0.2. After sequencing, the readings were filtered by the PGM software to remove the polyclonal (homopolymers > 6) and low-quality sequences (quality score ≤ 20).

### 2.6. Taxonomic Assignment and Bacterial Diversity

Amplicon Sequence Variants (ASV) were determined from reads that met the quality criteria using the QIIME2-2022.2 pipeline [32]. Representative sequences were taxonomically annotated with Silva 138 database with the weighted pre-trained classifier (Weighted Silva 138, 99% OTUs full-length sequences) [33]. Further analyses were performed with R 4.2.1 [34] into RStudio 2022.07.01 + 554 IDE [35]. Data were imported into R with qiime2R 0.99.6 package [36], phyloseq 1.40.0 package [37] was used for the analysis of microbial communities with relative abundances. For intra-sample diversity Chao1, Shannon, Simpson, InvSimpson, ACE, and Fisher indexes were calculated. Analysis of the inter-sample diversity was carried out with UniFrac distance, and Non-Metric Multidimensional Scaling (NMDS) ordination with vegan 2.6.2 package [38]. Core microbiota heat map (50% prevalence, 1% detection) and Spearman’s rank correlation of bacteria with variables (anthropometric, clinic, dietary, and SCFA content) were elaborated with microbiome 1.18.0 [39] and ComplexHeatmap 2.12.1 [40] packages. Differential abundance analysis was performed with DESeq2 1.36.0 [41]. Data were managed with tydiverse 1.3.2 [42]. Correlograms were made with psych 2.2.5 package [43]. Figures were elaborated with ggplotify 0.1.0 [44], ggpubr 0.4.0, RColorBrewer 1.1.3 [45], and pals 1.7 [46]. To predict metabolic profiles of the bacterial metagenome from the sequencing data, PICRUSt v2 program was used, with the MetaCyc metabolic pathway database option. Statistical, taxonomic, and functional analysis software was used (STAMP v2.1.3) to determine significant differences in the functional metabolic pathways of the bacterial metagenome [47]. The pipeline script for analysis was included in the Appendix A.

### 2.7. Analysis of Short-Chain Fatty Acids by HPLC

The SCFA were measured from freeze-dried fecal samples using the Perkin Elmer-Flexar HPLC system (Waltham, MA, USA). Samples were pre-treated before injection in the equipment as follows: 0.5 mL deionized water was added to 50 mg of dehydrated sample, then 100 µL of concentrated HCl, and mixed by vortex for 15 s. Subsequently, 1 mL of ethyl ether was added and mixed on an orbital shaker (speed 80 rpm for 20 min). A centrifugation step was applied for 5 min at 3500 rpm, recovering the supernatant and repeating the ether extraction step. Finally, 500 µL of 1M NaOH was added to the final supernatant, taking the aqueous phase and filtering with a 0.45 µm PTFE filter. After filtering, 100 µL of concentrated HCl was added and mixed by vortex for 6 s.

The mobile phase used consisted of two solutions: 80% of solution A, composed of 20 mM KH_2_PO_4_ (J.T. Baker, State of Mexico, Mexico) at pH 2.2 (adjusted with phosphoric acid J.T. Baker, State of Mexico, Mexico), and 20% of solution B, composed of acetonitrile (J.T. Baker, State of Mexico, Mexico), as previously described [48]. A C18 Discovery^®^ 10 cm × 2.1 mm, 5 µm particle size column was used (Cat. #569220-U, Supelco^®^, Sigma-Aldrich, St. Louis, MO, USA). The detection threshold for the method was 0.85 mM/L for formic acid, 0.80 mM/L for acetic acid, 1.00 mM/L for propionic acid, 0.08 mM/L for butyric acid, and 2.00 mM/L for valeric acid. All chromatographic data were processed using Chromera (v4.1.2.6410)—HPLC Flexar Software (PerkinElmer, Waltham, MA, USA).

### 2.8. Analysis of Metabolites by ESI FT-ICR MS

Solarix XR (Bruker, Bremen, Germany) Fourier Transform Ion Cyclotron Resonance Mass Spectrophotometer (FT-ICR MS) was calibrated in positive and negative Electrospray (ESI) mode with sodium trifluoroacetate solution. Samples were processed as described for the HPLC methods and injected into the instrument with a Hamilton 250 µL syringe at 120 µL/h flow rate by positive and negative ESI (450 V, 1 nA Capillary; −500 V, 9.451 nA End Plate Offset) to ensure optimal ionization efficiency and a larger number of identified metabolites. The acquisition conditions were as follows: 42.99 Low *m*/*z*, 3000 High *m*/*z*, 24 Average scans, 0.1 Accum (s), and 8M resolution. The source gas tune was N_2_, at 1 bar nebulization, 2 L/min dry gas, and 176.5 °C dry temperature. The DataAnalysis v.6.0. program was used for the generated data. The name and structure of the candidate of metabolites were assigned using Bruker Compass MetaboScape 2022 b v.9.0.1. For the statistical analysis, OriginPro 2021 v.9.8.0.200 was used.

### 2.9. Sequence Accession Numbers

The sequence FASTQ files and corresponding mapping files for all samples used in this study were deposited in the NCBI repository BioProject PRJNA884382 https://www.ncbi.nlm.nih.gov/sra/PRJNA884382 (accessed on 10 October 2022).

## 3. Results

### 3.1. Characteristics of Mothers in the Sample

Results depicted in Table 1 show that women in the sample had an average age of 28 years, with a gestational age of approximately 32 weeks. The anthropometry indicated a height of 1.57 m, which is normal among Mexican women, and a tendency of higher weight for women in the GD, PD, and PE groups compared to CO women. BMI data showed more than 68% of women were overweight or obese in the groups. The blood test revealed that women of the GD, PD, and PE groups had higher levels of fasting glucose and triglycerides. The average parity was <3 births among the 54 studied women. Most of the women had a secondary education level and were in free unions, being housewives as their main activity (Table 1). The measurement of SCFA in feces had no statistical difference in formic, acetic, propionic, butyric, and valeric acid concentration among the studied groups, with formic acid being the most abundant (Table 1).

The analysis of the nutrimental information collected from the participants revealed significant differences for nine macronutrients among the CO, GD, PD, and PE groups (Table 2); however, only statistically significant differences for energy, carbohydrates, protein, total fiber, cereal, and sodium intake were observed for CO vs. GD, after applying a Benjamini–Hochberg post-hoc test (Appendix A).

### 3.2. Alfa and Beta Diversity of the Gut Microbiota in Gestational Health Conditions

The gut microbiota diversity was inferred by characterization of the fecal microbiota using V3−16S rRNA gene libraries and high-throughput DNA sequencing. Five million total reads were obtained, with an average of 87,000 reads/sample and a median quality score of 32 (Table 3). The sequencing was satisfactory as shown in the rarefaction plots (Appendix A). Analyses characterizing the alfa diversity in samples in CO, GD, PD, and PE groups (Figure 1A), did not show a statistically significant difference for the Chao1, ACE, Shannon, Simpson, InvSimpson, and Fisher indexes (Appendix A). Additionally, the evaluation of the beta diversity, showed that only the microbiota diversity in CO and GD differed with statistical significance (*p* = 0.01) (Figure 1B).

### 3.3. Diversity of the Fecal Microbiota Shows a Predominance of Proteobacteria Phylum

When the microbiota diversity was evaluated at the phylum level in CO, GD, PD, and PE groups, a higher abundance of Proteobacteria was observed, followed by Firmicutes and Bacteroidota in all groups (Figure 1C). There was, however, no statistically significant difference for these phyla among the groups (Appendix A). At the genus level, *Sphingomonas* (Proteobacteria) was the most abundant taxa for CO (12.32%) and PD (21.35%); the genus *Blautia* (Firmicutes) for GD (17.27%), and the genus *Enterococcus* (Firmicutes) for PE (14.99%) (Figure 2A). However, there were only statistically significant differences between CO and GD groups for *Achromobacter*, *Allorhizobium-Neorhizobium-Pararhizobium-Rhizobium*, *Mesorhizobium* (Proteobacteria), *Bifidobacterium*, and *Cutibacterium* (Actinobacteriota) (Appendix A). The bacterial taxa, whose abundance contrasted when comparing CO versus PD, were *Allorhizobium-Neorhizobium-Pararhizobium-Rhizobium*, *Methylobacterium-Methylorubrum*, *Pseudomonas* (Proteobacteria), and *Bifidobacterium* (Actinobacteriota) (Appendix A). On the other hand, there was only a statistically significant difference for *Corynebacterium* (Actinobacteriota), *Mesorhizobium* (Proteobacteria), and *Streptococcus* (Firmicutes) when comparing the abundances between CO and PE (Appendix A).

The bacterial diversity was also explored in a core microbiota model assessment, where the abundance of taxa with >1% reads presented in at least 50% of the samples was comparatively analyzed and the results were plotted in a heat map of relative abundances normalized for the core taxa of each group (Figure 2B). As observed in the heat map, the CO group had comparatively more abundance of *Mesorhizobium* (Proteobacteria), *Alistipes*, Bacteroides (Bacteroidota), and *NK4A136* (Firmicutes) and less abundance of *Methylobacterium* (Proteobacteria), *Enterococcus*, *Gemella*, *Finegoldia*, *Staphylococcus*, and *Streptococcus* (Firmicutes), than the other experimental groups (Figure 2B). The GD group had more abundance of *Paraclostridium*, *Lactobacillus* (Firmicutes), *UCG-001* (family Prevotellaceae, Bacteroidota), *Bosea*, *Escherichia* (Proteobacteria), and less abundance of *Cutibacterium*, *Micrococcus* (Actinobacteriota), *Variovorax*, *Achromobacter* (Proteobacteria), and *Alistipes* (Bacteroidota) than CO, PD, and PE (Figure 2B). In the PE group, the abundance of *Corynebacterium* (Actinobacteriota), *Methylobacterium*, *Paracoccus*, *Acinetobacter* (Proteobacteria), *Muribaculaceae* (Bacteroidota), *Helicobacter* (Campilobacterota), *Enterococcus*, *Gemella*, *Finegoldia*, and *Staphylococcus* (Firmicutes) was increased and only the abundance of *Reyranella* (Proteobacteria) was comparatively decreased with respect to the abundance in the other groups (Figure 2B). Finally, in the PD group, the abundance of *Mycobacterium* (Actinobacteriota)*, Sphingopyxis, Pseudomonas, Stenotrophomonas, Variovorax*, *Allorhizobium*, and *Sphingomonas* (Proteobacteria) was increased and the abundance of *Bradyrhizobium* (Proteobacteria)*, Muribaculaceae* (Bacteroidota)*, Lactobacillus* (Firmicutes)*, Bifidobacterium* (Actinobacteriota), and *Enterococcus* (Firmicutes) was decreased.

### 3.4. DESeq2 Analysis Reveals the Abundance of Taxa Characterized by the Phylum Firmicutes

The comparative analysis DESeq2 using the ASV table revealed that bacterial diversity in GD was characterized by increased abundance of *UCG-014* (Class Clostridia), *Staphylococcus*, *Clostridium_sensu_sctricto_1, Hungatella* (Firmicutes), *Rothia* (Actinobacteriota), *Enterobacter* (Proteobacteria), and decreased abundance of a different ASV of *Pseudomonas* (Proteobacteria), *Hungatella* (Firmicutes), *Obscuribacteraceae* (Cyanobacteria), and *Bacteroides* (Bacteroidota) in comparison to the CO group (Figure 3A), (Appendix A). The bacterial diversity of PD was characterized by an increased abundance of *Pseudomonas* (Proteobacteria) and decreased abundance of an uncultured Firmicutes, *UCG-002*, *UBA1819* (the last three family Oscillospiraceae), *NK4A136* (family Lachnospiraceae), *Lacnoclostridium*, *Hungatella*, *Fusicatenibacter*, *Faecalibacterium*, *Enterococcus*, *Dorea*, *Clostridium_sensu_sctricto_1*, *Blautia*, *Bifidobacterium* from Actinobacteriota phylum, and *Bacteroides* (Bacteroidota) (Figure 3B), for CO, (Appendix A). On the other hand, the diversity in PE exhibited an increased abundance of *Staphylococcus* and *Gemella* (Firmicutes), while there was a decrease in the abundance of members of the Firmicutes phylum like uncultured ASV, *UCG-002, NK4A136* (the last three family Oscillospiraceae), *Paraclostridium*, *Hungatella*, halli_group (genus *Eubacterium*), *Faecalibacterium*, *Dorea*, *Clostridium_sensu_sctricto_1*, *Blautia* and *Obscuribacteraceae* (Cyanobacteria), *Mycobacterium* (Actinobacteriota), *Iamia* (Actinobacteriota), and *Bacteroides* (Bacteroidota) in comparison to the CO group (Figure 3C), (Appendix A).

### 3.5. Spearman’s Correlation Analyses of Selected Metadata with Bacterial Abundance

The Spearman correlation analyses using the metadata and ASV files detected positive and negative correlations for the explored variables. Relevant results for CO and GD were obtained when the bacterial abundance was correlated with anthropometric and biochemical data. For the case of the CO group, there was a positive correlation of members of the phylum Firmicutes with gestational age (*Enterococcus*), total cholesterol (*Clostridium*); triglycerides (*Enterococcus*, *Gemella*, *Streptococcus*, *vadinBB60*, Class Clostridia, *Clostridium sensu stricto 1*; phylum Proteobacteria with weight (*Achromobacter*), BMI (*Reyranella*), body surface (*Achromobacter*), total cholesterol (*Reyranella*, *Mesorhizobium*, *Sphingopyxis*); phylum Actinobacteriota (*Mycobacterium*, *Microbacterium*, *Lawsonella*, *Rothia*), and phylum Cyanobacteria with heart rate (Obscuribacteraceae). In contrast, there was a negative correlation between members of phylum Campilobacterota (*Helicobacter*) with gestational age, total cholesterol, Actinobacteriota (*Mycobacterium*) with size, and Bacteroidota (Muribaculaceae) with gestational age (Figure 4A). On the other hand, the GD group had only a positive correlation for members of the phylum Actinobacteriota (*Microbacterium*, *Cutibacterium*), and Firmicutes (*Bacillus*) with age; Bacteroidota (*UCG-001*, family Prevotellaceae) and Campilobacterota (*Helicobacter*) with fasting glucose, and Proteobacteria (*Enterobacter*) with triglycerides (Figure 4B).

The correlation analysis of dietary data with bacterial diversity data for the CO group disclosed positive correlations of phylum Actinobacteriota (*Iamia*) with saturated and monosaturated fatty acids; phylum Firmicutes with sodium, vegetable use (*Enterococcus*), sucrose, fruits and berries use (*vadinBB6*, Class Clostridia); phylum Bacteroidota with fructose (Muribaculaceae), glucose and starch (*Odoribacter*) and glucose (Muribaculaceae), and phylum Proteobacteria (*Reyranella*) with sodium. There was a negative correlation of Firmicutes with energy intake, fat intake, cholesterol, cereal, milk products, saturated fatty acids (*Gemella*), cholesterol, sour milk products, protein intake (*Clostridium sensu stricto 1*), polyunsaturated fatty acids (*NK4A136*, family Lachnospiraceae), milk products (*vadinBB60*, Class Clostridia), and cholesterol (*Paraclostridium*); phylum Actinobacteriota with fat intake, saturated fatty acids, monosaturated fatty acids, and sodium (*Rothia*), and phylum Proteobacteria with protein intake, cholesterol, and saturated fatty acids (*Escherichia*) (Figure 4C). In the case of the GD group, a positive correlation was observed for members of phylum Bacteroidota with protein intake, monosaturated fatty acids, meat (*Bacteroides*), and Proteobacteria with cereal (*Enterobacter*). A negative correlation was found for members of phylum Proteobacteria with polyunsaturated fatty acids, starch, glucose, fructose (*Mesorhizobium*), with sodium, starch (*Bosea*), sour milk products (*Achromobacter*), sucrose (*Escherichia*), fructose (*Sphingomonas*); phylum Firmicutes with energy intake (*Clostridium sensu stricto 1*), sodium (*Staphylococcus*), and caffeine (*Enterococcus*); phylum Actinobacteriota with starch, and fructose (*Microbacterium*), milk products (*Rothia*) and fructose (*Cutibacterium*), and for phylum Bacteroidota with starch, fruits, and berries used, fructose, glucose and sodium (*UCG-001*, family Prevotellaceae) (Figure 4D).

In the case of the correlation analysis of CO with SCFA, there was a positive correlation of Proteobacteria phylum members with fumaric acid (*Mesorhizobium*, *Sphingopyxis*); phylum Firmicutes with fumaric acid (*Staphylococcus*), propionic acid (*Staphylococcus*), and phylum Actinobacteriota with fumaric acid (*Kocuria*). A negative statistically significant correlation was only found for phylum Proteobacteria with propionic acid (*Variovorax*) (Figure 4E). For GD, there were three positive correlations of the phylum Firmicutes with butyric acid (*Enterococcus*, *Streptococcus*), and propionic acid (*Lactococcus*); while a negative correlation was found for the phylum Proteobacteria with fumaric acid (*Enterobacter*), with acetic acid (*Achromobacter*, *Bosea*, *Pseudomonas*, *Undibacterium*, *Bradyrhizobium*, *Sphingomonas*), propionic acid (*Sphingomonas*), butyric acid (*Sphingomonas*), and valeric acid (*Bosea*, *Undibacterium*); phylum Firmicutes with valeric acid (*Staphylococcus*); phylum Actinobacteria with acetic acid (*Microbacterium*, *Cutibacterium*), propionic acid (*Microbacterium*, *Cutibacterium*), and butyric acid (*Cutibacterium*); phylum Bacteroidota with butyric acid (Muribaculaceae), with valeric acid, acetic acid, propionic acid, and total SCFA (*UCG-001*, family Prevotellaceae) (Figure 4F).

### 3.6. Prediction of Bacterial Metagenome and Metabolite Profile in Fecal Samples

The PICRUSt analysis of the ASV table determined a prediction metagenome and identified interesting functional metabolic pathways in the bacterial microbiota, where the mean proportion (%) of each metabolic pathway contrasted among the studied groups after a strict statistical analysis (Welch’s test, with Bonferroni correction). There were thirteen metabolic pathways when comparing CO and GD; being most of them catabolic and primarily detected in CO bacterial microbiota (Figure 5A), (Appendix A). For the case of CO versus PD, twenty-seven pathways were reported by the analysis, of which fourteen were more abundant in CO, five anabolic and nine catabolic, and thirteen in PD, being eight anabolic and five catabolic (Figure 5B), (Appendix A). Finally, the comparative analysis between CO and PE revealed only one catabolic pathway for vitamin B6 degradation in CO (Figure 5C), (Appendix A).

The metabolite profile in fecal samples collected from CO, GD, PD, and PE groups, was explored by FT-ICR MS. The profile analysis of identified metabolites using positive ionization for CO versus GD (Figure 6A), CO versus PD (Figure 6B), CO versus PE (Figure 6C) and negative ionization mode for CO versus GD (Figure 6D), CO versus PD (Figure 6E), and CO versus PE (Figure 6F) did not show a clear clustering of samples under comparison. However interesting metabolites like trioxopyrrolopyridine, 9,9’-spirobi[carbazol-9-ium], and complex phenolic, valeric, arachidic, and capric acids among others were identified under positive (Appendix A) as well as negative (Appendix A) ionizations.

## 4. Discussion

Characterization of the gut microbiota diversity associated with gestational health conditions is of great importance to understand the effect of changes in the microbiota and host interactions during the development of a new human being. Unlike other reports, in our study, Proteobacteria had the highest relative abundance, followed by Firmicutes and Actinobacteria [28]. The gut of healthy humans is dominated by four major bacterial phyla: Firmicutes, Bacteroidetes, and to a lesser degree, Proteobacteria and Actinobacteria [4,49]. It has been reported that gut microbiota changes remarkably from the first to the third trimester during a healthy pregnancy, increasing diversity and reducing richness, with an increased abundance of Proteobacteria and Actinobacteria [28].

A previous review reported changes in the gut microbiota composition in gestational diabetes pregnancies in comparison with normoglycemic pregnancies, where alpha diversity was decreased and beta diversity increased. The variations in the gut microbial composition during pregnancy showed an increased Proteobacteria/Actinobacteria ratio, and an increase in Firmicutes and Bacteroidota abundance was observed as well [50]. A study reported similar diversity and community structure in women with gestational diabetes compared to control women concerning Observed OTUs, Shannon’s diversity index, and Pielou’s evenness index [27]. These results are similar to those obtained in our study.

In our GD group, the abundance of *Achromobacter*, *Rhizobium*, *Bifidobacterium*, and *Mesorhizobium* is decreased. We also found more abundance of *UGC*-*014*, *Clostridium_sensu_stricto_1* (class Clostridia), *Staphylococcus*, *Bosea*, *Rothia*, and *Enterobacter*. *Bifidobacterium* is a known primary colonizer of the intestinal epithelium and producer of SCFA [51]. The lower abundance of this genus induces the downregulation of GLP-2 synthesis, a protein involved in the regulation of gut barrier function [52]. *Bifidobacterium* is reported highly abundant in Crohn’s disease with respect to the control group in a study in Canadian population [53]. Women in Denmark in the third trimester with gestational diabetes, diagnosed by oral glucose tolerance test, showed an increased abundance of phylum Actinobacteria and genera *Collinsella*, *Rothia*, and *Desulfovibrio* compared with the normoglycemic group [27].

Regarding *Clostridium sensu stricto 1*, the bacterial high density of this genus may cause epithelial intestinal inflammation. Certain *Clostridium* spp. are harmful to host health, for instance, epithelial inflammation observed in weaned piglets may be correlated with *Clostridium sensu stricto 1* enrichment in their intestinal mucosa [54]. Additionally, the correlation of the expression of pro-inflammatory cytokines, such as IL-1β and TNF-α with colon inflammation caused by *Clostridium sensu stricto 1*, has been observed [55]. The presence of this genus correlated inversely with the consumption of cholesterol, protein intake, and sour milk products in the CO group, and GD group, as well as associated with energy intake, in our work. *Clostridium* has been implicated in the maintenance of mucosal homeostasis and prevention of inflammatory bowel disease and with an increase in the anti-inflammatory activity of Treg lymphocytes in mice, therefore *Clostridium* might modulate various aspects of the immune system [56]. In the GD group, we found a correlation between the presence of genus *UCG-001* (family Prevotellaceae) with fasting glucose; other member of the same family *Prevotella*, produces SCFA increasing incretin secretion and reducing inflammation and insulin resistance [56].

Through several mechanisms, gut microbial dysbiosis can contribute to the development of proteinuria, a strong risk factor for the development and progression of chronic kidney disease, hypertension, and diabetes, in addition to preeclampsia [57]. In the PE group of our study, the Firmicutes to Bacteroidota ratio was increased, as reported for hypertensive subjects, in another study [58]. Our results for the PE group, are similar to a report on pregnant Chinese women, where there are no significant differences in diversity between the pre-eclampsia and control groups [59]. This work in Chinese women, also reported that the relative abundance of Proteobacteria decreased significantly in the control group, and the relative abundance of Firmicutes was significantly lower in the pre-eclampsia group than in the control group; in contrast, in our work we found a tendency to increase in the PE group.

In the PE group, some genera increased (*Bosea*, *Escherichia*, *Staphylococcus*, *Enterococcus*), while others decreased (*Sphingomonas*, *Microbacterium*, *Pseudomonas*, *Bifidobacterium*, and *Lactobacillus*) in comparison with the CO group. In patients with proteinuria-associated diseases, a reduction in the abundance of *Lactobacillus* and *Bifidobacterium* species has been reported. These two genera are among the most well-known probiotics with important functions such as protection of the gut barrier structure, SCFA, nitric oxide, and vitamin complex production [57,60]. Other genera were observed to increase, such as *Corynebacterium*, *Methylobacterium*-*Methylorubrum*, and *Streptococcus* in contrast, *Mesorhizobium* was diminished with a significant difference in the PE group of our work. The genus *Mesorhizobium* belongs to the Proteobacteria phylum; this genus consists of 51 species, isolated mostly from root nodules of various leguminous plants [61]. Some strains of *Mesorhizobium* can oxidate acids (i.e., acetic acid), as well as assimilate sugars, in addition, to being an important nitrogen fixer in legume roots [62]. *Methylobacterium* species are opportunistic pathogens in immunocompromised patients, described as a cause of cross-contaminations, that frequently colonizes in the hospital setting and are major inhabitants of aqueous environments, including potable water supplies and hospital tap water, and some *Methylobacterium* infections have been associated with raw vegetable consumption [63]. Finally, with the use of antibiotics, decreased incidence of cases of pre-eclampsia was demonstrated (Chinese population) in patients with hypertension, where decreased microbial richness and diversity, and overgrowth of bacteria such as *Prevotella* and *Klebsiella* were observed [58].

In the PE group, of our work, *Gemella* and *Staphylococcus* (Firmicutes) are two taxa with differential abundance (according to DESeq2). *Gemella* is a common resident of mucosal membranes, with a high abundance in cases of Crohn’s disease and ulcerative colitis [53]. We found that *Gemella* was positively correlated with triglycerides. The increased abundance of this bacteria was considered a risk factor in pregnancies with overweight and metabolic disease according to one report on obese Italian adults [64,65]. *Streptococcus* was found increased in the PE group, the abundance of this bacteria has been reported to be higher in numerous inflammatory diseases [53] and alterations in the prevalence of these bacteria may alter the vascular tone and contribute to the development of hypertension and pre-eclampsia [66].

In the PD group of our work, Proteobacteria increased and the Firmicutes phyla decreased along with other taxa. Some genera belonging to the Rhizobiaceae family detected in the PD group, are known as potential nitrogen-fixing symbionts of legumes, isolated from root nodules [67]. Other studies report that genera like *Rhizobium* are found as contaminants of DNA extraction and PCR kits, and this is also the case for *Methylobacterium*-*Methylorubrum* [68]. Results in relative abundance at the genus level, show that although the abundance of some taxa did not show a significant difference among groups, they are still important since they are associated with changes occurring in pregnancy. For example, *Prevotella* (Bacteroidota) and *Clostridium* (Firmicutes) display different changes in some diseases such as hypertension and diabetes (Type 1 or 2) [58]. *Pseudomonas* was found differentially abundant in our work in patients with PD. This genus has been reported as an opportunistic pathogen associated with mice suffering from diabetes mellitus [69].

Concerning SCFA production, in the GD group, a direct correlation between the genus *Lactococcus* and the presence of propionic acid was observed. In a study conducted in mice, the administration of a food supplement was associated with an increase in *Lactococcus* and other microbial genera, and greater production of SCFA, including propionic acid, as reported in our work [70]. In another study, a correlation of this genus with propionic acid originating from the biotransformation of L-threonine and L-methionine was observed [71]. A positive correlation of the genus *Streptococcus* with butyric acid was also observed in patients with GD in our group. Other studies have shown that species of this genus, such as *Streptococcus mitis*, are capable of oxidizing butyric acid mainly to acetic acid mainly [19], and this could be a mechanism of regulation and compensation of acetic acid since many genera in GD were observed to have an inverse correlation with acetic acid, genera belonging mostly to the phylum Proteobacteria.

A previous study found that the metabolic pathways related to the intestinal microbiota in patients with gestational diabetes are different from those in healthy female controls [72]; in this study, it was further shown that the amino acid content in fecal samples was decreased in patients with gestational diabetes. In our work, we observed an overall decrease in the metabolic pathways involved with the metabolism of amino acids such as tryptophan, alanine, aspartate, glutamate, cysteine, and methionine. It has been observed that, in patients with gestational diabetes, there is an increase in serum levels of branched amino acids, such as isoleucine, tyrosine, and alanine [73]. *Lactobacillus* and *Bacteroides* are bacteria related to amino acid metabolism (especially tryptophan), found with decreased abundance in the GD group of our work. In germ-free mice colonized with *Lactobacillus* and *Bacteroides*, it was found that a particular *Lactobacillus reuteri*, was able to promote the production of double-positive intraepithelial lymphocytes (DP IEL) [74]. DP IEL cells are present in the small intestine, normally helping the body to tolerate food components and other foreign molecules, attenuating immune responses. The importance of this study in germ-free mice is that these bacteria needed tryptophan to promote the production of this type of host cells, with a role in reducing low-grade inflammation, which is common in patients with gestational diabetes. We observed metabolic pathways related to the use of carbohydrates, previously reported for gestational diabetes. In general, affected women have a decrease in the correct assimilation of carbohydrates obtained through the diet [23], which is related to dysbiosis in the gut microbiota, since there is a decrease in microorganisms related to the use of carbohydrates and an increase in bacterial species related to insulin resistance (*Akkermansia*) and glucose intolerance (*Blautia*) [27]. Concerning the differences in metabolic pathways found in the CO and PD groups, it was possible to appreciate an increase of pathways related to LPS synthesis in the PD group, in general, type II diabetes mellitus is associated with obesity and consumption of high-fat diets, and this, in turn, causes an increase in intestinal permeability caused by high serum levels of LPS, favoring the characteristic low-grade inflammation of these patients [50].

The comparison of metabolic pathways between the CO and PE groups revealed an increase in Vitamin B6 degradation in the CO group. The intestinal microbiota metabolism provides the host with many nutrients including amino acids and B-complex vitamins including Vitamin B6, important cofactors for carbohydrate metabolism and DNA synthesis. A large amount of B-vitamins are then obtained from the diet or intestinal microbiota. Vitamin B6 metabolism has been associated with bacteria, such as *Bacteroides*, *Feacalibacterium*, *E. coli*, *Klebsiella*, and *Salmonella*, among others [75], and a decrease in *Bacteroides* abundance was found in the PE group of our work.

The metabolome analysis did not show differential metabolites among our studied groups. There are few studies where candidate metabolite biomarkers for gestational diabetes were evaluated. A review on this topic reports changes in free fatty acids (FFAs), branched-chain amino acids (BCAAs), lipids, and organooxygen compounds, which differentiated both the control and gestational diabetes [76]. However, most of these studies analyzed the metabolome of serum samples, while there are few studies of metabolome performed on fecal samples from patients with gestational diabetes. For instance, in a study, the metabolome was evaluated with ^1^H-NMR, from fecal samples of pregnant women with gestational diabetes and control groups, at 24–28 gestational weeks. Here, a clear clusterization of metabolites between both groups was observed and five biomarker metabolites for gestational diabetes were also proposed [77]. Further metabolomic studies with more samples are needed to identify the specific microbial metabolites and pathways involved in diabetic onset and pathology. The results obtained in our work suggest that disturbances of the gut microbiota contribute to the occurrence of GD, PD, and PE.

## 5. Conclusions

In this work, we find fecal microbial profiles, with predictive metagenomes associated with different gestational health conditions, such as GD, PD, and PE, in Mexican women. Although a major limitation of this work is the low number of samples, the results and conclusions are valid for the studied participants.

## Figures and Tables

**Figure 1 nutrients-14-04818-f001:**
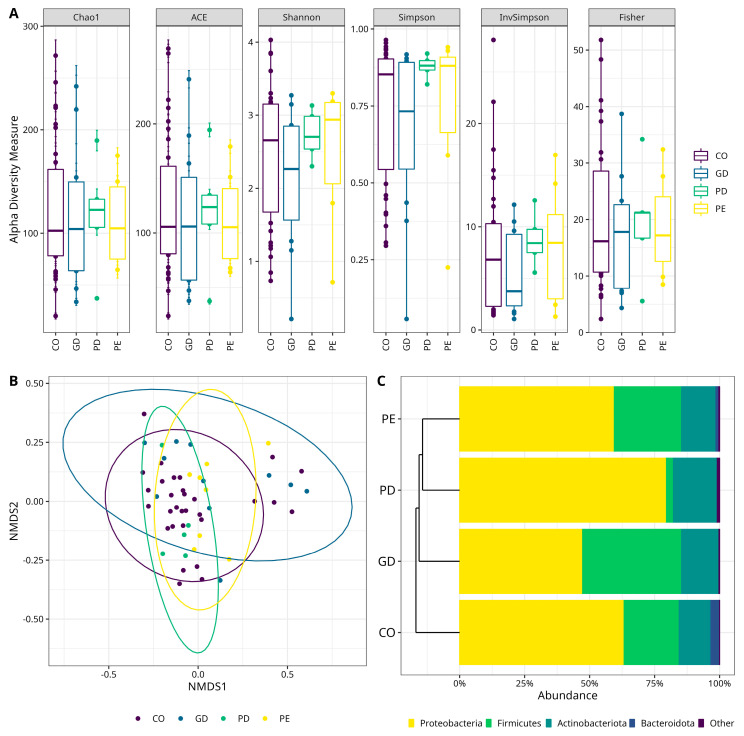
Characterization of the bacterial diversity in the different studied samples. (**A**) Alpha diversity box plots. The Y-axes indicate the values for the species richness indexes (Chao1, ACE), and diversity indexes (Shannon, Simpson, InvSimpson, and Fisher). The type of sample is shown on the right. Appendix A—(see Appendix A for numerical data of indexes). (**B**) Beta diversity Non-Metric Multidimensional Scaling (NMDS) scatter plots. The graphics show bacterial beta diversity calculated by NMDS ordination based on the UniFrac distance matrix. The scatter plots were generated in R. The samples CO and GD differed significantly according to ANOSIM (*p* = 0.01). (**C**) Bacterial Phyla relative abundance stacked bar plots. Color sectors indicate taxa as denoted by tags at the bottom of the figure; abundances are shown as a percentage on the X-axis. The type of sample is shown on the left side of the figure. The graphic shows the four top more abundant phyla, while “Other” includes phyla with <1% relative abundance—(see Appendix A for numerical data abundances and statistical test for CO versus GD, Appendix A; CO versus PD, Appendix A, and CO versus PE, Appendix A). PE (Pre-Eclampsia), PD (Pre-gestational Diabetes), GD (Gestational Diabetes), and CO (Control).

**Figure 2 nutrients-14-04818-f002:**
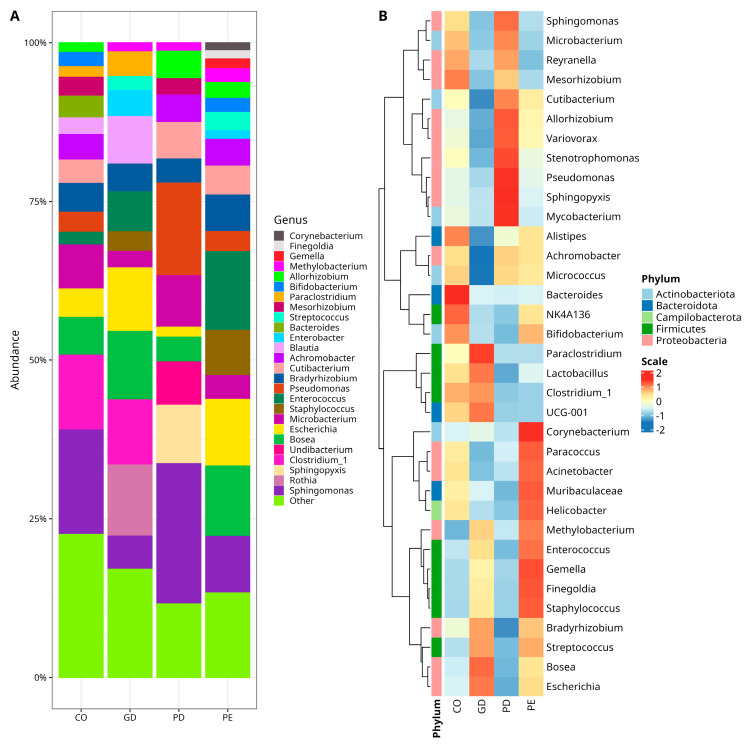
Relative abundances of bacterial genera in the studied samples. (**A**) Stacked bar plots with relative abundances of bacteria. Color sectors indicate taxa as indicated by tags at the right side of the figure; abundances are shown as percentages on the Y-axis. The graphic shows the twenty-six topmost abundant genera, while “Other” group genera with <1% relative abundance—(see Appendix A for numerical data abundance and statistical test for CO versus GD, Appendix A; CO versus PD, Appendix A, and CO versus PE, Appendix A). (**B**) Core microbiota heatmap among samples. Columns show the abundance of core microbiota members with a prevalence of at least 50% in the samples and an abundance ≥1%. The color scale from blue (−2) to red (2) indicates the relative abundance normalized from the core taxa of groups. Color keys for phyla are shown on the left side of the figure. *NK4A136* (Lachnospiraceae), *UCG-001* (Prevotellaceae), *Escherichia* (*Escherichia-Shigella*), *Allorhizobium* (*Allorhizobium-Neorhizobium-Pararhizobium-Rhizobium*), *Methylobacterium* (*Methylobacterium-Methylorubrum*), *Clostridium_1* (*Clostridium_sensu_stricto_1*). The type of sample is indicated at the bottom of the figure, where PE (Pre-Eclampsia), PD (Pre-gestational Diabetes), GD (Gestational Diabetes), and CO (Control).

**Figure 3 nutrients-14-04818-f003:**
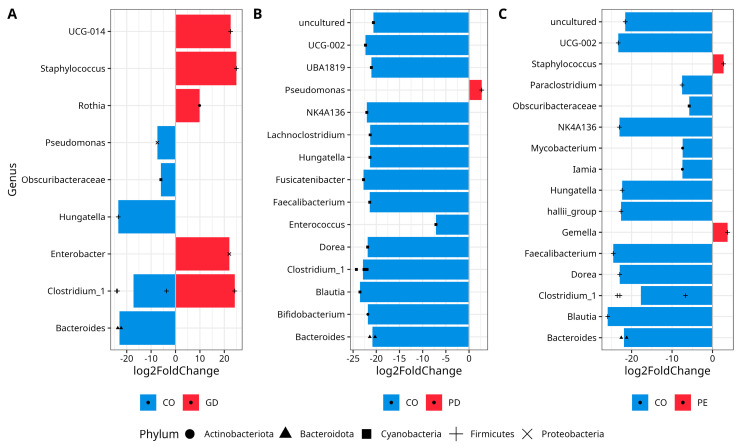
Differential abundance analysis of bacterial genera with DESeq2. The Figure shows data for (**A**) CO vs. GD, (**B**) CO vs. PD, and (**C**) CO vs. PE. Normalization was made with size factor from counts geometric means and the Wald test was applied to calculate differences between groups, False Discovery Rate (FDR) was used to correct *p*-values. Log2 Fold Change is shown by the horizontal bars. Bacterial taxa with *q* values <0.05 are written alongside the Y-axis. Phyla are shown with a black solid circle (Actinobacteriota), black solid triangle (Bacteroidota), black solid square (Cyanobacteria), plus symbol (Firmicutes), and cross symbol (Proteobacteria) at the bottom. The repetition of symbols indicates that more than one ASV was reported in the analyses. *UCG-014* (Class Clostridia), *Clostridium_1* (*Clostridium_sensu_stricto_1*), uncultured (Oscillospiraceae), *UCG-002* (Oscillospiraceae), *UBA1819* (Oscillospiraceae), *NK4A136* (Lachnospiraceae), *halli_group* (Eubacterium). —(see Appendix A for full taxon description, log2FoldChange, *p*, and *p-adjusted* values for CO versus GD, Appendix A; CO versus PDF, Appendix A, and CO versus PE, Appendix A).

**Figure 4 nutrients-14-04818-f004:**
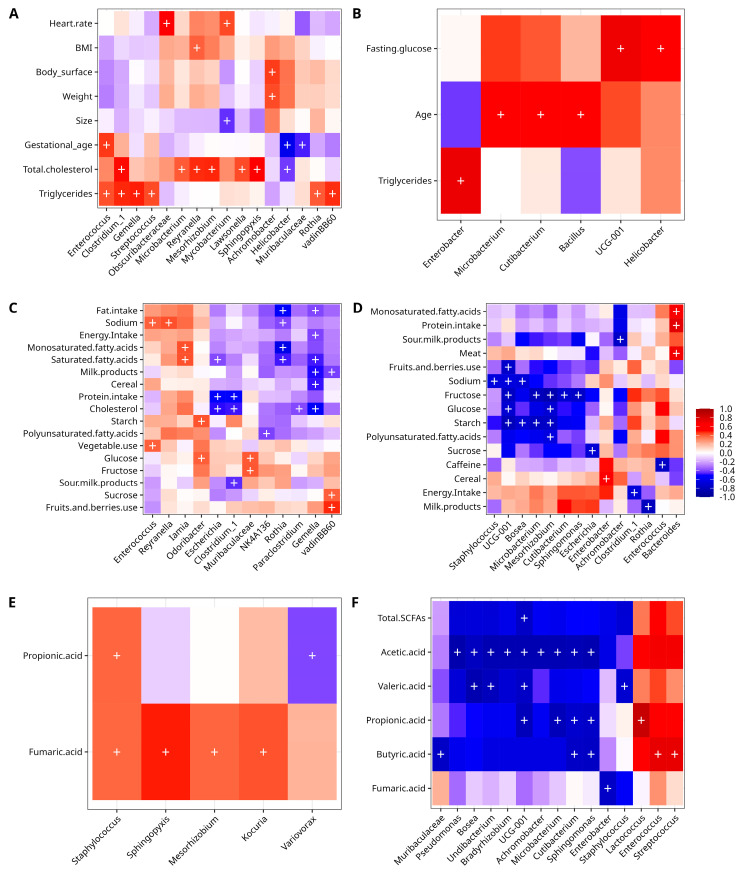
Spearman correlation analysis of clinical data and other variables with bacterial abundance. Anthropometric and clinical for CO (**A**), and GD (**B**); dietary for CO (**C**) and GD (**D**), and SCFA for CO (**E**) and GD (**F**). Columns in the heatmaps show the bacterial taxa, while rows show the numerical metadata. The correlation is measured by the color key from blue (−1, negative) to red (+1, positive). The plus symbol “+” denotes a significance of *p* < 0.001. *vadinBB60* (Class Clostridia); *UCG001* (Prevotellaceae); *NK4A136* (Lachnospiraceae), *Clostridium_1* (*Clostridium_sensu_stricto_1*), *Escherichia* (*Escherichia-Shigella*).

**Figure 5 nutrients-14-04818-f005:**
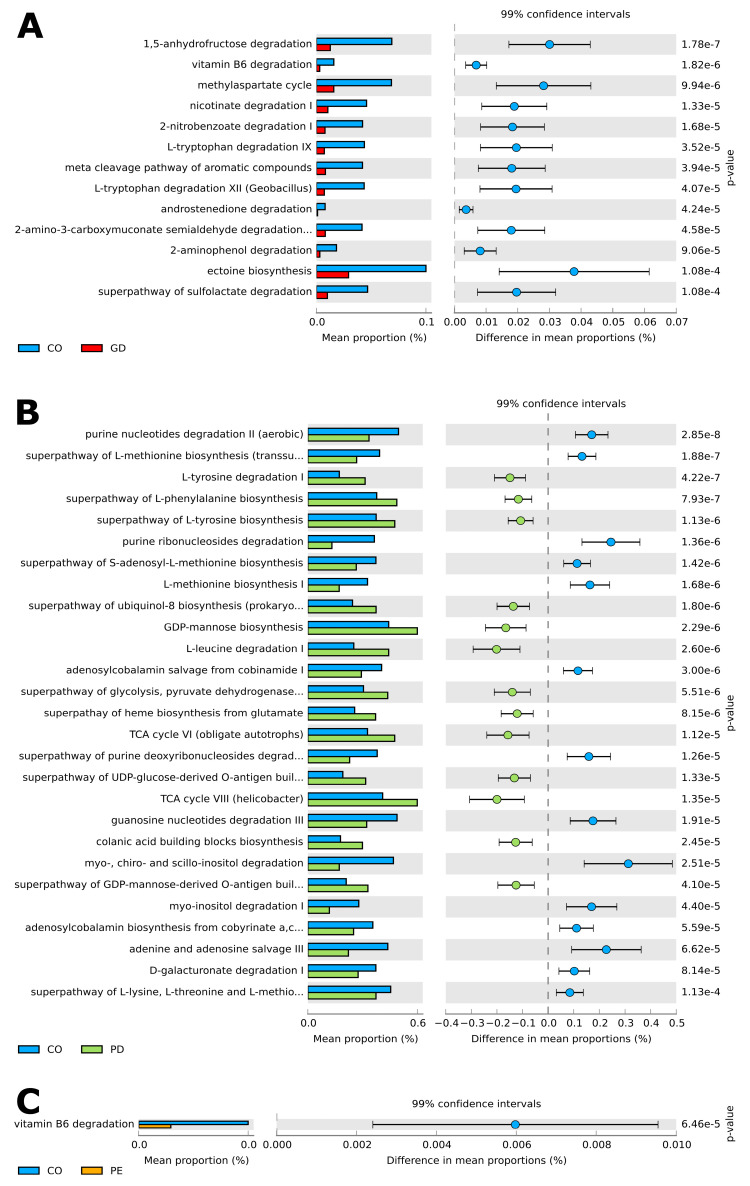
Prediction of functional microbial metabolic pathways using PICRUSt 2 analysis with the MetaCyc database. The graphics show the abundance of (**A**) thirteen statistically significant metabolic pathways between CO (blue color) and GD (red color) bacterial communities. (**B**) Twenty-seven statistically significant metabolic pathways between CO (blue color) and PD (green color) bacterial communities; and (**C**) one statistically significant metabolic pathway between CO (blue color) and PE (orange color) bacterial communities. Confidence intervals are indicated on top, while the mean proportions and differences in mean proportions with percentage scale are shown underneath each graphic. Groups are identified by a tab placed below the graphics. A Welch test was applied with a Bonferroni post-hoc. Corrected *p*-values are shown on the right side of each graphic. —(see Appendix A for all included statistically significant pathways *q* < 0.05).

**Figure 6 nutrients-14-04818-f006:**
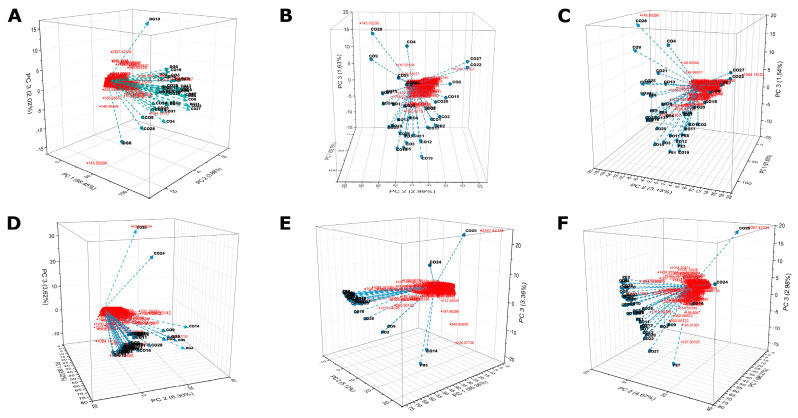
Analysis of metabolites in fecal samples by FT-ICR MS. Metabolites extracted from fecal samples were analyzed using a Fourier Transform Ion Cyclotron Resonance Mass Spectrophotometer (Solarix XR Bruker) calibrated in positive mode for CO versus GD (**A**), CO versus PD (**B**), CO versus PE (**C**), and negative mode for CO versus GD (**D**), CO versus PD (**E**), and CO versus PE (**F**) see Materials and Methods. For each graphic, the PC1, PC2, and PC3 axes indicate the PCA ordination as a percentage of the total variance explained. The red color is the m/z values and the lines with conic tips represent the samples that are identified by tags.

**Table 1 nutrients-14-04818-t001:** General data for participants of the study by groups.

Variable		CO	GD	PD	PE	*p*-Value
Number of subjects (*n* = 54)		30	11	5	8	nd
Age (years)		24.70 (±6.59)	31.09 (±3.60)	28.20 (±5.26)	28.13 (±7.99)	0.0097 *
Gestational age (weeks)		32.38 (±5.99)	33.05 (±5.62)	29.85 (±4.60)	33.37 (±3.51)	0.5947
Anthropometry	Height (m)	1.55 (±0.08)	1.56 (±0.07)	1.60 (±0.06)	1.57 (±0.06)	0.2474
	Weight (Kg)	67.59 (±11.87)	69.28 (±10.56)	75.28 (±6.77)	74.09 (±13.51)	0.1085
	Normal (BMI 18.5–24.9)	10 (33.33%)	3 (27.27%)	2 (40.00%)	2 (25.00%)	-
	Overweight (BMI 25.0–29.9)	8 (26.67%)	3 (27.27%)	1 (20.00%)	3 (37.50%)	-
	Obesity (BMI > 30.0)	12 (36.67%)	5 (45.45%)	2 (20.00%)	3 (12.50%)	-
	Body Surface Area (m^2^)	1.66 (±0.16)	1.69 (±0.14)	1.78 (±0.12)	1.81 (±0.10)	0.1350
	Heart rate (beats/min)	83.55 (±12.01)	82.64 (±6.93)	81.20 (±13.57)	97.63 (±14.87)	0.0828
Blood test	Fasting glucose (mg/dL)	80.21 (±10.08)	83.89 (±9.43)	115.00 (±14.54)	86.25 (±16.07)	0.0007 *
	Total Cholesterol (mg/dL)	218.08 (±64.05)	241.42 (±87.29)	174.25 (±13.82)	229.25 (±79.72)	0.1670
	Triglycerides (mg/dL)	234.17 (±88.71)	256.96 (±100.78)	456.25 (±258.78)	349.75 (±142.23)	0.0297 *
	Glycosylated hemoglobin (%)	nd	5.85 (±1.15)	9.53 (±4.79)	Nd	-
Risk factors ^#^	Alcoholism	1 (3.33%)	0 (0.00%)	0 (0.00%)	0 (0.00%)	-
	Smoking	2 (6.67%)	0 (0.00%)	1 (20.00%)	0 (0.00%)	-
	Drug addiction	1 (3.33%)	0 (0.00%)	0 (0.00%)	0 (0.00%)	-
Average parities	Total	2.63 (±1.96)	3.00 (±1.48)	3.00 (±1.22)	2.13 (±1.13)	0.4747
	Vaginal	0.70 (±1.26)	0.18 (±0.40)	0.20 (±0.45)	0.50 (±1.07)	0.4543
	Cesarean	0.70 (±0.84)	1.09 (±0.83)	0.40 (±0.55)	1.13 (±0.64)	0.1559
	Abortions	0.40 (±0.67)	0.45 (±0.69)	0.80 (±0.45)	0.25 (±0.46)	0.2585
Socioeconomic data						
Educational level	Primary school (6 years)	3 (10.00%)	0 (0.00%)	0 (0.00%)	0 (0.00%)	-
	Secondary school (3 years)	12 (40.00%)	6 (54.55%)	3 (60.00%)	4 (50.00%)	-
	High school (3 years)	9 (30.00%)	3 (27.27%)	2 (40.00%)	4 (50.00%)	-
	University school (4–5 years)	6 (20.00%)	2 (18.18%)	0 (0.00%)	0 (0.00%)	-
Marital status	Free union	14 (46.67%)	7 (63.64%)	2 (40.00%)	6 (75.00%)	-
	Married	7 (23.33%)	4 (36.36%)	2 (40.00%)	2 (25.00%)	-
	Single	9 (30.00%)	0 (0.00%)	1 (20.00%)	0 (0.00%)	-
Main activity	Housewife	23 (76.67%)	10 (90.91%)	5 (100.00%)	7 (87.50%)	-
	General employee	4 (13.33%)	1 (9.09%)	0 (0.00%)	0 (0.00%)	-
	Student	3 (10.00%)	0 (0.00%)	0 (0.00%)	1 (12.50%)	-
Fecal SCFA	Formic acid	30.13 (±24.88)	21.77 (±6.32)	24.85 (±7.61)	19.37 (±4.54)	0.1207
(mM/100 mg)	Acetic acid	0.71 (±0.93)	0.90 (±1.45)	0.74 (±0.70)	0.43 (±0.76)	0.8657
	Propionic acid	1.13 (±2.11)	0.94 (±0.94)	0.57 (±0.78)	0.71 (±1.22)	0.8678
	Butyric acid	0.29 (±0.56)	0.11 (±0.23)	0.16 (±0.36)	<0.08 **	0.3378
	Valeric acid	3.53 (±3.32)	2.72 (±2.27)	6.36 (±8.63)	0.95 (±1.62)	0.1518

CO, Controls; GD, Gestational Diabetes; PD, Pre-gestational Diabetes; PE, Pre-Eclampsia; BMI, Body Mass Index; m, meters; Kg, kilograms; mg, milligrams; L, liters; mM, millimolar; mg; milligrams; SCFA, Short Chain Fatty Acids; **, less than the detection limit of 0.08 mMol/L; Standard deviation is shown as ± values; *p*-value was calculated Kruskal–Wallis test; *p* < 0.05 are considered statistically significant differences.; nd, not determined; -, *p*-value was not calculated since it was a categorical variable. *, indicates statistically significant value. ^#^, the number corresponds to the subject with this classification.

**Table 2 nutrients-14-04818-t002:** Nutrimental data for participants in the study by groups.

Macronutrients	CO	GD	PD	PE	*p*-Value
Number of subjects	26	10	4	6	nd
Energy Intake (kcal/day)	2496.0 (±1214.0)	1143.3 (±529.0)	2562.0 (±1706.0)	1892.0 (±828.3)	0.0103 *
Fat intake (g/day)	74.9 (±55.4)	39.0 (±27.2)	77.0 (±102.1)	58.6 (±49.2)	0.3292
Carbohydrates intake (g/day)	255.3 (±157.8)	121.4 (±80.4)	150.7 (±190.8)	158.4 (±110.9)	0.0480 *
Protein intake (g/day)	124.5 (±85.6)	46.3 (±23.5)	76.5 (±53.5)	82.7 (±50.5)	0.0243 *
Total fiber intake (g/day)	21.0 (±10.6)	10.2 (±5.6)	16.7 (±10.0)	16.2 (±8.1)	0.0170 *
Cholesterol (g/day)	62.1 (± 56.4)	34.2 (±32.5)	36.9 (±27.70)	68.2 (±81.7)	0.6719
Saturated fatty acids (g/day)	19.0 (±13.4)	10.0 (±8.3)	9.0 (±4.302)	15.7 (±16.0)	0.3267
Monosaturated fatty acids (g/day)	13.3 (± 9.8)	6.4 (±5.8)	5.3 (±1.8)	7.3 (±4.6)	0.1260
Polyunsaturated fatty acids (g/day)	5.2 (±7.9)	1.6 (±1.37)	1.4 (±1.0)	2.3 (±1.7)	0.0505
Starch (g/day)	27.3 (±28.8)	11.1 (±20.7)	1.9 (±0.8)	29.0 (±29.4)	0.0290 *
Vegetable use (kcal/day)	210.5 (±380.5)	31.4 (±21.3)	465.1 (±857.7)	156.2 (±319.7)	0.1477
Fruits and berries use (kcal/day)	207.6 (±264.8)	34.5 (±38.5)	25.2 (±17.5)	41.4 (±27.6)	0.0550
Cereal (kcal/day)	1018.0 (±443.5)	552.2 (±257.0)	1009.0 (±514.0)	1035.0 (±456.5)	0.0150 *
Milk products (kcal/day)	222.6 (±274.0)	150.0 (±178.7)	101.3 (±181.0)	136.8 (±184.1)	0.4649
Sour milk products (kcal/day)	112.4 (±186.3)	18.6 (±16.1)	77.6 (±68.5)	20.0 (±36.6)	0.1176
Meat (kcal/day)	163.4 (±192.3)	103.0 (±144.0)	77.9 (±79.6)	124.4 (±191.2)	0.9181
Sucrose (g/day)	29.0 (±35.4)	4.9 (±4.5)	6.6 (±3.9)	5.0 (±5.7)	0.0303 *
Fructose (g/day)	7.8 (±8.5)	2.3 (±2.6)	1.1 (±0.2)	3.1 (±2.9)	0.0501
Glucose (g/day)	4.5 (±4.0)	1.8 (±2.3)	0.7 (±0.2)	2.4 (±2.6)	0.0303 *
Caffeine (mg/day)	40.0 (±74.5)	38.3 (±62.9)	79.3 (±88.7)	52.2 (±80.4)	0.5784
Sodium (mg/day)	1426.0 (±891.4)	494.8 (±278.7)	1225.0 (±1515)	1589.0 (±1600.0)	0.0093 *

Kruskal–Wallis test, significant data with * *p* < 0.05. The standard deviation is shown as ± within the parentheses. nd, not determined.—see Appendix A for macronutrients with statistical significance between groups, numerical data abundances, and statistical test for CO versus GD, after Benjamini–Hochberg post-hoc test, Appendix A).

**Table 3 nutrients-14-04818-t003:** Ion torrent semiconductor DNA sequencing summary (*n* = 54).

**Parameter before Trimming**		
	Forward reads total	4,711,327
	Forward reads mean	87,246.80
	Min-Max forward reads	5247–568,088
	Sequence length (median)	173 nt
	Samples with <10,000 reads	8
**Parameter after trimming**		
	QS (median)	32
	Total ASV counts	2849
	Identified ASVs	2668

QS, Quality Score; Trimmed less than 170; ASVs, Amplicon Sequencing Variants; *n*, number of samples; Percentage of identity (97%); nt, nucleotide.

## Data Availability

The sequence FASTQ files and corresponding mapping files for all samples used in this study were deposited in the NCBI repository BioProject PRJNA884382 https://www.ncbi.nlm.nih.gov/sra/PRJNA884382 (accessed on 10 October 2022).

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
