# Peer review of "Gut Microbiota Associated with Gestational Health Conditions in a Sample of Mexican Women"

_nutrients, 2022, doi:10.3390/nu14224818_

Round 1

Reviewer 1 Report

In the study titled "Gut micobiota associated with gestational health conditions in a sample of Mexican woman" the authors conducted an observational, retrospective, case control study where fecal microbiota with predictive metagenomes were compared between healthy control and woman with gestational diabetes, preeclampsia and pre-pregnancy diabetes. The manuscript is well written with only a editing needed. The methods are sufficiently defined and results clearly described. A major concern is the low samples numbers in some of the groups. However, the authors mention this in the conclusion. Overall the tables and figures are appropriate. Only a few minor edits are required. 

Line 37: Remove massive (Remove throughout the manuscript)

Line 56: Change sentence to "The functional contribution of                                                  the microbiota is important as  many chronic human diseases, including obesity [3],  "

Line 67: Change "Nowadays" to a more appropriate word/phrase.

Line 111: Change "premium" to "important".

Line 137: Change sentence "It is important to mention that sample collections occurred from July to October 2021"

Line 148: rewrite the following sentence "A food frequency questionnaire was also applied to the patient"

Line 150: Add information about stool collection. When and how where stool samples collected? Were they self-collected by participants? How were the samples stored prior to DNA extraction. 

Line 237: Rewrite sentence " For the massive data was used DataAnalysis"

Line 251-254: Change mothers to women.

Table 1: Add explanation of what numbers for Risk factors refer to.

Table 1: Add unit to Fecal SCFA

Line 554: Rewrite sentence "In patients with proteinuria-associated diseases, it has been reported a reduction in the abundance of Lactobacillus..."

Author Response

Manuscript ID: ijms-2001080

Title: Gut microbiota associated with gestational health conditions in a sample of Mexican Women

Authors: Tizziani Benítez-Guerrero, Juan Manuel Vélez-Ixta, Carmen Josefina Juárez-Castelán, Karina Corona-Cervantes, Alberto Piña-Escobedo, Helga Martínez-Corona, Amapola De Sales-Millán, Yair Cruz-Narváez, Carlos Yamel Gómez-Cruz, Tito Ramírez-Lozada, Gustavo Acosta-Altamirano, Mónica Sierra-Martínez, Paola Berenice Zárate-Segura and Jaime García-Mena

Reviewer 1 (Round 1)

Comments and Suggestions for Authors

In the study titled "Gut micobiota associated with gestational health conditions in a sample of Mexican woman" the authors conducted an observational, retrospective, case control study where fecal microbiota with predictive metagenomes were compared between healthy control and woman with gestational diabetes, preeclampsia and pre-pregnancy diabetes. The manuscript is well written with only a editing needed. The methods are sufficiently defined and results clearly described. A major concern is the low samples numbers in some of the groups. However, the authors mention this in the conclusion. Overall the tables and figures are appropriate. Only a few minor edits are required.

Line 37: Remove massive (Remove throughout the manuscript)

Answer: The word massive was replace by “High-throughput DNA sequencing”, former line 37.

Former line 171, the information was amended.

Former line 237, “massive” was changed to “generated” in the context of current line 236.

Former line 277, “massive semiconductor DNA sequencing” was changed to “High-throughput DNA sequencing” in the context of current line 278.

Line 56: Change sentence to "The functional contribution of the microbiota is important as many chronic human diseases, including obesity [3],"

Answer: the original sentence in former line 56 “The functional contribution of the microbiota is so relevant, that many chronic human diseases, including obesity [3],” was changed to "The functional contribution of the microbiota is important as many chronic human diseases, including obesity [3]," in current lines 57-58 as requested.

Line 67: Change "Nowadays" to a more appropriate word/phrase.

Answer: “Nowadays” was changed to “The prevalence of GD” in current line 68.

Line 111: Change "premium" to "important".

Answer: as the reviewer requested “premium” was changed to “important” in current line 113.

Line 137: Change sentence "It is important to mention that sample collections occurred from July to October 2021"

Answer: the sentence “It is important to mention that samples were from July to October 2021,” was changed to “It is important to mention that all sample collection occurred from July to October 2021” as requested, current line 139-140.

Line 148: rewrite the following sentence "A food frequency questionnaire was also applied to the patient".

Answer: the sentence “A food frequency questionnaire was also applied to the patient” was amended and now it reads “For each patient, a food frequency questionnaire, designed to obtain information about eating habits was applied”, see current lines 151-152.

Line 150: Add information about stool collection. When and how where stool samples collected? Were they self-collected by participants? How were the samples stored prior to DNA extraction.

Answer: we thank the reviewer for getting to our attention this issue. The requested information about stool sample collection is informed in line 140 for when, and line 122-122 for where. The indication that stools samples were provided by the participants and stored at -70° C until further use is indicated in line 143-144.

Line 237: Rewrite sentence "For the massive data was used DataAnalysis"

Answer: the sentence “For the massive data was used DataAnalysis” was changed to “The DataAnalysis v.6.0 program was used for the generated data” in line 235-236.

Line 251-254: Change mothers to women.

Answer: the requested change was made in current lines 247 and 251.

Table 1: Add explanation of what numbers for Risk factors refer to.

Answer: on regard of this issue, a convenient annotation indicated by number sign symbol (#), was added in “Risk factors” variable name, and the footnote of Table 1.

Table 1: Add unit to Fecal SCFA

Answer: on regard of this issue, the units (mM/100mg) were added to the “Fecal SCFA” variable name, and the footnote of Table 1.

Line 554: Rewrite sentence "In patients with proteinuria-associated diseases, it has been reported a reduction in the abundance of Lactobacillus..."

Answer: the sentence was rewritten as requested, and now it reads “In patients with proteinuria-associated diseases, a reduction in the abundance of Lactobacillus and Bifidobacterium species has been reported” lines 551-553.

---end-of-text---

Reviewer 2 Report

The manuscript is of good quality, deals with a sound issue of selected gestational pathologies related to gut microbiota diversities. The study was designed approprietaly although moderate number of patients was involved. Nevertheless, robust data were collected and extended data analysis was presented. The Authors correctly applied the methodology, results are cleary presented and illustrated.

Some criticism is related to the Discussion which demands more linguistic efforts, as the conclusions disappear in many too long sentences:

page 17 line 595, "In relation..."

page 17 line 602, "A positive..."

page 18 line 617, "In a study..."

page 18 line 625, "Our work..."

page 18 line 638, vitamin B-6 vs. Vitamin B6

page 19 line 659 "The finding..."

Author Response

Manuscript ID: ijms-2001080

Title: Gut microbiota associated with gestational health conditions in a sample of Mexican Women

Authors: Tizziani Benítez-Guerrero, Juan Manuel Vélez-Ixta, Carmen Josefina Juárez-Castelán, Karina Corona-Cervantes, Alberto Piña-Escobedo, Helga Martínez-Corona, Amapola De Sales-Millán, Yair Cruz-Narváez, Carlos Yamel Gómez-Cruz, Tito Ramírez-Lozada, Gustavo Acosta-Altamirano, Mónica Sierra-Martínez, Paola Berenice Zárate-Segura and Jaime García-Mena

Reviewer 2 (Round 1)

Comments and Suggestions for Authors

The manuscript is of good quality, deals with a sound issue of selected gestational pathologies related to gut microbiota diversities. The study was designed approprietaly although moderate number of patients was involved. Nevertheless, robust data were collected and extended data analysis was presented. The Authors correctly applied the methodology, results are cleary presented and illustrated.

Some criticism is related to the Discussion which demands more linguistic efforts, as the conclusions disappear in many too long sentences:

Answer: we thank the reviewer for this constructive comment. For clarity in the narrative, the whole Discussion was thoroughly reviewed and rewritten.

page 17 line 595, "In relation..."

Answer: the word was changed to “Concerning SCFAs production…” in current line 592.

page 17 line 602, "A positive..."

Answer: a modification in the paragraph was made in lines 598 and 599, and the phrase “A positive correlation” was kept emphasizing the positive correlation found by the analysis.

page 18 line 617, "In a study..."

Answer: “In a study” was changed to “In another report…” current line 613.

page 18 line 625, "Our work..."

Answer: The sentence “Our work also demonstrated alteration..” was changed to “We observed metabolic pathways…” current line 620.

page 18 line 638, vitamin B-6 vs. Vitamin B6

Answer: “vitamin B-6” was updated to “Vitamin B6” in current lines 633 and 635.

page 19 line 659 "The finding..."

Answer: “The results” substituted “The findings” in current line 652.

---end-of-text---